# Effects of a Traction Device for Head Weight Reduction and Neutral Alignment during Sedentary Visual Display Terminal (VDT) Work on Postural Alignment, Muscle Properties, Hemodynamics, Preference, and Working Memory Performance

**DOI:** 10.3390/ijerph192114254

**Published:** 2022-10-31

**Authors:** Ju-Yeon Jung, Hwi-Young Cho, Chang-Ki Kang

**Affiliations:** 1Department of Health Science, Gachon University Graduate School, Gachon University, Incheon 21936, Korea; 2Department of Physical Therapy, College of Health Science, Gachon University, Incheon 21936, Korea; 3Neuroscience Research Institute, Gachon University, Incheon 21565, Korea; 4Department of Radiological Science, College of Health Science, Gachon University, Incheon 21936, Korea

**Keywords:** VDT workstation, VDT postural alignment, muscle stiffness and tone, blood velocity, VADS, 2-back working memory

## Abstract

The forward head posture of visual display terminal (VDT) users induces various physical and cognitive clinical symptoms. However, few studies have been conducted to identify and solve problems associated with VDT posture. This study aimed to examine the adverse effects of VDT posture and the positive effects of traction-combined workstations by measuring postural alignment, muscle properties, blood velocity, preference, and working memory. Thirty-four healthy VDT users (18 males and 16 females aged 20–30 years) participated in the experiment at three workstations, including conventional (VDT_C), head support (VDT_S), and upright (VDT_U) workstations. They conducted 2-back working memory task. The craniovertebral angle (CVA), muscle tone and stiffness, blood velocity and visual analogue discomfort scale (VADS) were measured to examine the influence of workstations. VDT_C showed increased muscle tone or stiffness in the levator scapulae (LS), suboccipital muscle (SM), and sternocleidomastoid muscle (SCM) and an increased reaction time (RT) in working memory. However, VDT_S showed decreased stiffness and tone of SM and improved comfort. In addition, VDT_U showed decreased stiffness or tone of the LS and SCM and improved blood velocity and RT. In conclusion, maintaining neutral alignment significantly improved working memory performance, muscle properties, and blood velocity.

## 1. Introduction

As non-contact online services are growing, work with mobile devices, including computers, has become an essential competency for everyone, and the incidence of symptoms or diseases related to the usage of computers and mobile devices is substantially increasing owing to the increased use time and number of users. 

Visual display terminal (VDT) syndrome is a representative clinical symptom caused by the long-term operation of display terminals that causes serious health problems. In particular, VDT users, due to prolonged sedentary behavior, are at an increased risk of chronic diseases, such as cardiovascular disease and diabetes [1,2]. They are also at an increased risk of muscular disorders such as myalgia and disc degeneration because of the increased stress of soft tissue and spine due to poor posture, such as forward head and kyphotic thoracic posture [3,4,5,6]. In addition, forward head posture, which is closely related to neck pain, induces hypertonic contraction of the upper trapezius and sternocleidomastoid muscles [7,8,9,10]. 

Furthermore, poor posture can lead to impaired blood circulation around the neck and shoulders. For example, abnormal cervical lordosis reduces blood flow in the vertebral artery [11]. It is also known to cause complicated problems related to increased stress and poor cognitive and psychological performance [11,12,13,14]. Subjects with slouched postures tend to be lethargic, have decreased levels of arousal, and experience increased stress and depression [12,13,14]. Therefore, poor VDT posture can adversely affect the work performance of VDT workers owing to decreased physical, physiological, and behavioral performance. However, few studies have identified VDT posture problems using physical, physiological, and cognitive functions simultaneously. Therefore, there is a need to investigate the adverse effects of VDT posture by measuring changes in postural alignment, muscle properties, hemodynamics, cognitive function, and preferences.

VDT users want to maintain an upright posture. However, it is still difficult to maintain neutral alignment during VDT work because their shoulders gradually slouch forward in the process of VDT work to view small display screens. More than 70% of VDT users work with slouched shoulders and a forward-bending posture [15]. Therefore, the VDT workstation is getting a lot of attention lately. However, despite general safety guidelines such as physical exercises included stretching, resistance exercise and cervical spine extension exercise to prevent VDT syndrome and musculoskeletal discomfort of VDT users [16,17,18], it is difficult for VDT users who work for a long time to follow the guidelines, and practical solutions to prevent VDT posture during work are lacking. 

Therefore, we propose a new method that can be effectively applied to VDT users for a long time by designing a VDT workstation using a traction device. In this study, two VDT workstations were designed. One aimed to reduce the head weight and the other to maintain an upright posture. In head weight support VDT workstation (VDT_S), one of the risk factors for VDT posture is musculoskeletal load by head weight. The forward head posture, such as the VDT posture, increases the twofold compression force, increases the fourfold anterior shear force in the cervical spine, and induces abnormal muscle hyper- and hypo-tonicity [19]. In addition, these abnormal changes can cause headache in VDT users [20,21,22]. Therefore, we aimed to apply the VDT_S to eliminate risk factors induced by head weight and examine its effect on postural alignment, muscle properties, hemodynamics, cognitive function, and preferences. The upright VDT workstation (VDT_U) maintains an upright posture by providing tactile feedback when VDT users are bent forwards during VDT work. The traction belt of VDT_U is located under the user’s chin so that users can naturally sense the pulled belt by forward bending. This effect has already been reviewed in a previous study, which reported that a forward-bent feedback device using tactile sensing significantly reduces VDT posture [23]. 

With new VDT workstations using a traction device as a novel solution to prevent the VDT posture, this study therefore aims to examine the adverse effects of VDT posture and advantages of VDT_S and VDT_U in physical, physiological, and cognitive aspects by measuring postural alignment, muscle properties, blood velocity, preferences, and working memory performance. The first hypothesis of this study is that the conventional VDT posture (VDT_C) affects muscle properties, working memory performance, and blood velocity. Second, VDT_S and VDT_U are effective in reducing the problems of VDT_C. Consequently, the present study aims to provide practical and sustainable solutions for VDT users.

## 2. Materials and Methods

### 2.1. Participants

This study examined 34 right-handed visual display terminal (VDT) users (18 males and 16 females aged 20–30 years) who had used a computer for more than 5 years and for at least 4 h a day [24]. All participants provided written informed consent to participate in the study. This study was approved by the institutional review board of Gachon University (IRB No. 1044396-202101-HR-015-01). The participants had no history of musculoskeletal, neurological, or psychiatric disorder. Subjects who had any head, neck, or upper extremity disorders that may have interfered with the VDT work performance and who had a cranio-vertebral angle (CVA) >50° during VDT work were excluded from the study to investigate habituated forward head posture [25,26,27]. The CVA is the angle between the line connecting the tragus to the distal tip of the spinous process of C7 and the horizontal line [26]. The general characteristics of the study participants are presented in Table 1. 

### 2.2. Experimental Protocol

Participants were explained about three different VDT workstations and practiced on the 2-back working memory task for more than 5 min to familiarize themselves with the task and the workstation. We performed experiments using VDT_C, VDT_S, and VDT_U in a random order to eliminate the order effect. The participants had an adaptation time of 5 min at the first workstation (ex, VDT_C) and then performed a 2-back task for 5 min, providing the reaction time (RT) and false rate. Immediately after completion of the 2-back task, the visual analogue discomfort scale (VADS) on the workstation was assessed. Then, the CVA, muscle properties and blood velocity were measured at the same workstation. The participants were asked to maintain their experimental posture until the end of the measurements. After all measurements were completed, the next workstation proceeded. Experiments in the next workstation were conducted after a 5 min break to wash out the effects of the previous workstation. The above procedure was equally applied in each workstation (VDT_C, VDT_S, and VDT_U). 

### 2.3. Intervention

Three VDT workstations were used in this study. VDT_S and VDT_U were created using a TM300 traction system (TM300 Traction System, ITO Co., Tokyo, Japan). The traction device was positioned vertically at the center of the subject’s head to effectively support the head weight. In the VDT_C, participants were asked to perform their ordinary posture, which was the habituated forward head posture in VDT users. The VDT works induce a forward bending posture toward the monitor. The forward head and trunk flexion become postural habit during VDT works, causing to lower cervical flexion, upper cervical extension, shoulder elevation and kyphotic thoracic posture [28,29,30]. In VDT_C condition, the participants were allowed to take adaptation time similar to pre-training before the start of the experiment, and they unconsciously took the habitual forward head posture. The participants were seated in an adjustable-height chair with a backrest, and their knee and hip joints were positioned at a 90° angle and feet on the floor [31]. The participants assumed a habituated forward head posture during the 2-back task. The forward-bent range was set according to individual preferences. Both hands and elbows were positioned on the keyboard and desk, respectively (Figure 1). The keyboard was positioned on the desk according to the participant’s preference after the desk height was set. An adjustable-height desktop monitor was used to adjust the vertically downward viewing angle within 10° and positioned 60 cm away horizontally from their eyes [32].

The VDT_S was used to reduce the head weight with a traction device in the VDT_C posture. The head weight was assumed to be 7.3% of the subjects’ total body weight [33]. To ensure head weight reduction, the subjects were asked to relax their neck extensor muscles and their heads were checked to determine whether they fell. In the VDT_U, participants were asked to maintain an upright posture facing forward without tilting their necks, with their backs on the backrest. A cervical traction belt was placed under the chin, without contact, to prevent forward bending. Therefore, only when subjects bent forward was the belt brought into contact with the chin so that the subjects’ posture change could be recognized, and the upright posture could be maintained.

### 2.4. Measurements

#### 2.4.1. Working Memory

A 2-back working memory task was performed to assess the concentration and working memory according to VDT workstations. A 2-back task is to distinguish positive (target) or negative (non-target) stimuli with the series of numbers for 5 min. For 2-back stimulation, one of nine different single digits (1–9) was randomly displayed for the 2-back stimulation in white on the middle of a black background and in 30 points font size. The digits were presented for 500 ms, followed by a cross mark for 1500 ms. The “M” key of keyboard was set as a positive response and “N” key was a negative response. In the 2-back task, a positive stimulus was defined as the one with a 2-steps back and any other digit as a negative stimulus. During the 2-back task, participants were asked to press “M” or “N” key according to the positive or negative stimulus. A total of 150 responses were recorded during 5 min of the 2-back task. The 2-back task was performed on each of the VDT_C, VDT_S and VDT_U workstations. Therefore, the three different 2-back tasks used in this study were programmed using DMDX software 2017 (University of Arizona, Tucson, AZ, USA). Participants were asked to respond as quickly and accurately as possible and were trained sufficiently before the experiment. 

#### 2.4.2. Postural Alignment Measurement

Changes in the cervical angle were evaluated with CVA, which can be measured with the intersection angle between the horizontal line and the line connecting the C7 vertebra and the tragus of the ear. CVA is a representative method for distinguishing forward head posture. The markers were attached to the anatomical landmarks of the C7 vertebra and tragus and photographed using a camera placed 1.5 m away at the height of the acromion [27,34]. The CVA values were measured using ImageJ analysis software (National Institutes of Health, Bethesda, MD, USA) [27,34].

#### 2.4.3. Muscle Properties

Superficial skeletal muscle properties, such as stiffness (N/m) and muscle tone (Hz), were measured with a handheld myotonometer (Myoton AS, Tallinn, Estonia). Both sides of the suboccipital muscles (SM), upper trapezius (UT), levator scapulae (LS), and sternocleidomastoid (SCM) muscles were measured. Muscle stiffness is the magnitude of the force against the displacement of the fascia tissue and indicates the level of resistance of the myofascial tissue to external forces. Muscle tone is the intrinsic tension of a muscle in its passive or resting state without any voluntary contraction [35]. The measurement of muscle properties was performed in a seated posture in a stable and relaxed state, and the average value of three repeated measures was collected.

The measurement points of SM, UT and SCM muscles were as follows. SM was marked between the middle of the C2 spinous process and the occiput. The point of SM was placed above the atlanto-occipital joint and included the rectus capitis posterior [21,36,37]. The measuring point of UT was assessed based on the midpoint of the horizontal line connecting the acromion to the spinous process of the seventh cervical vertebra [35]. LS was marked just above the superior angle of the scapula [34]. SCM was examined at a point located midway between the insertion to the anterior surface of the manubrium sterni and the mastoid process of the temporal bone [8]. The myometric method had high test–retest reliability (ICCs = 0.79–0.96) [38]. 

#### 2.4.4. Blood Velocity

The systolic peak velocity (PSV) and end-diastolic blood flow velocity (EDV) of both common carotid arteries (CCA) were measured using color Doppler ultrasound (Prosound F37, Hitachi Healthcare, Ltd., Twinsburg, OH, USA) using a 4 cm linear transducer (5–10 MHz, Hitachi Healthcare, USA). An experienced ultrasonographer conducted all velocity measurements. A 60° angle correction was used to obtain Doppler velocities. The mean of the three beats was used as PSV and EDV (Figure 2). 

#### 2.4.5. Preference and Discomfort

The Visual Analogue Discomfort Scale (VADS) was used to assess the discomfort of each VDT workstation [39,40]. The “0” labels “very comfortable” and “10” “extremely uncomfortable.” VADS was highly correlated with visual analogue score (r = 0.54, *p* < 0.001) [41]. Discomfort was rated by the participants immediately after they performed the task in each condition. 

### 2.5. Statistical Analysis

Statistical analysis was performed using Jamovi 2.2.5 [42]. One factorial repeated-measure ANOVA was used to test for mean difference (MD) in every outcome among the factors (workstations) with three levels (VDT_C, VDT_S, and VDT_U). A standard criterion of statistical significance (*p* < 0.05) was applied to all analyses. The false rate in the 2-back task was expressed as a percentage of the number of incorrect responses divided by the total number of responses. The mean RTs were calculated from a total of 150 RTs measured at each workstation. Post hoc pairwise comparisons were conducted using Tukey’s honest significant difference test. The tables only show statistically significant outcomes.

## 3. Results

### 3.1. 2-Back Task Performance Changes According to VDT Workstations 

The traction workstation significantly changed the RT during the 2-back task (*F* = 5.32; *p* = 0.007). The VDT_U showed the fastest RT of 640.45 ms, which was significantly faster than VDT_C by 33.48 ms (4.97%) (*t* = 2.64; *p* = 0.033) and VDT_S by 42.75 ms (6.26%) (*t* = 2.52; *p* = 0.044) (Table 2). However, there was no significant difference in the false rate of the VDT workstation (*F* = 2.11; *p* = 0.129).

### 3.2. Quantitative Changes According to VDT Workstations

#### 3.2.1. CVA

The traction workstation had a significant effect on the CVA improvement during the 2-back task (*F* = 115.30; *p* < 0.001). CVA improved in both VDT_S and VDT_U compared to VDT_C, and the greatest improvement was observed in VDT_U. By comparing the mean difference (MD) of CVA, VDT_U was improved by 12.90° (*t* = −10.53, *p* < 0.001), and VDT_S was significantly improved by 2.50° compared to VDT_C (*t* = −2.75, *p* = 0.026) (Table 3). 

#### 3.2.2. Muscle Properties

Significant changes in muscle properties were also observed in the traction workstation. The muscle tone and stiffness of the LS for the scapula elevator differed significantly depending on the workstation. On average, the left and right tones in LS were the lowest in VDT_U and highest in VDT_S. Right tone in LS (Tone_R_LS) of VDT_U was significantly decreased by 1.15 Hz than VDT_S (*t* = 3.91; *p* = 0.001), and left tone in LS (Tone_L_LS) was significantly decreased in VDT_U than in VDT_S and VDT_C (VDT_C–VDT_U, MD = 0.59, *p* = 0.027; VDT_S–VDT_U, MD = 1.39, *p* < 0.001). Furthermore, only the left stiffness in LS (Stiffness_L_LS) showed a significant difference. Stiffness_L_LS was significantly higher in VDT_S than that in VDT_C and VDT_U (VDT_C–VDT_S, MD = −25.15, *p* < 0.001; VDT_S–VDT_U, MD = 34.09, *p* < 0.001) (Table 3).

In SM, which plays the role of erection and extension of the head, the tone and stiffness of the right and left SM were significantly changed by the traction workstation. The tone of the right and left SM (Tone_R_SM, Tone_L_SM) was significantly lower in VDT_S than in VDT_C (VDT_C–VDT_S in Tone_R_SM, MD = 1.06, *p* < 0.001; VDT_C–VDT_S in Tone_L_SM, MD = 0.81, *p* = 0.017). The stiffness of right and left SM (Stiffness_R_SM, Stiffness_L_SM) showed significant differences by the traction workstation (Stiffness_R_SM, *F* = 6.08, *p* = 0.004; Stiffness_L_SM, *F* = 5.28, *p* = 0.007), and both sides of SM stiffness were significantly decreased in VDT_S compared to VDT_C and VDT_U. Stiffness_R_SM of VDT_S was significantly decreased by 28.71 N/m compared to VDT_C (*p* = 0.002), and 26.15 N/m than VDT_U (*p* = 0.016). Stiffness_L_SM was significantly decreased by 25.41 N/m and 21.59 N/m compared to VDT_C (*p* = 0.012) and VDT_U (*p* = 0.016), respectively. In SCM, which plays the role of rotation and flexion of the head, stiffness was significantly decreased in VDT_U. The stiffness in right SCM (Stiffness_R_SCM) was significantly decreased by 11.38 N/m in VDT_U compared to VDT_C (*t* = 2.946, *p* = 0.016) (Table 3). However, there were no significant differences in the tone and stiffness of the UT at the VDT workstation. 

#### 3.2.3. Blood Velocity of CCA

The traction workstation had a significant effect on the right CCA PSV (R_PSV) (*F* = 4.80, *p* = 0.012). The average R_PSVs for VDT_U, VDT_S, and VDT_C were 116.40 cm/s, 109.82 cm/s, and 113.57 cm/s, respectively. Among them, the VDT_U significantly increased 6.58 cm/s compared to VDT_S (*p* = 0.029) (Table 3). However, EDV showed no significant differences at the workstation. 

### 3.3. Changes of Preferences According to VDT Workstations

The VADS was significantly affected by the traction workstation (*F* = 4.48; *p* = 0.015). The average VDT_S had the lowest VADS score (VDT_S, mean = 3.82; VDT_U, mean = 4; VDT_C, mean = 4.94), and when compared to VDT_C, a significant improvement of comfort was observed in VDT_S (*t* = 3.29; *p* = 0.007) (Table 4).

## 4. Discussion

We examined the adverse effects of VDT_C on physical, physiological, and cognitive function and further investigated the benefits of traction combined with VDT workstations (VDT_S and VDT_U). In this study, significant improvements in postural alignment, muscle properties, blood velocity, preference, and working memory performance were observed in the VDT_S and VDT_U groups. 

### 4.1. 2-Back Task Performance 

To the best of our knowledge, this is the first evidence that a forward-bent VDT posture is associated with poor performance on working memory tasks. We observed a significant improvement in RT in VDT_U compared with VDT_C and VDT_S (Table 2), probably due to the arousal effect of the upright posture. Similar to the present results, various effects of upright posture on cognitive function have been reported in previous studies. Wilkes (2017) and Cohen (2016) argued that a straight thoracic spine was associated with high arousal, and that forward head posture was associated with poor performance in episodic memory [12,14]. Therefore, the upright posture of VDT_U could respond faster to the 2-back task with the same false rate, which corresponds to an improvement in the working memory performance.

### 4.2. Postural Alignment 

In this study, the level of forward posture was evaluated by CVA, and VDT_U had a significant effect on maintaining an upright posture during VDT work (Table 3). The CVA (54.52°) after VDT work in VDT_U was within the normal range, indicating that the tactile feedback using the traction device worked effectively to help VDT users maintain an upright posture. This result was similar to that of a previous study in which a contact feedback device using biofeedback had a significant effect on minimizing postural changes [23]. In addition, in an ordinary VDT workstation (VDT_C), it was difficult for VDT users to maintain an upright posture for 5 min during VDT work [23]. However, VDT_U allowed VDT users to easily maintain an upright posture during 2-back task of 5 min. In addition, as studied in a previous study, the improved CVA is believed to help prevent the musculoskeletal and physiological problems of VDT users [43].

### 4.3. Muscle Properties

The upright posture using VDT_U significantly affected the mechanical properties of the muscle. Tone_R_LS, Tone_L_LS, Stiffness_L_LS, and Stiffness_R_SCM were significantly decreased in VDT_U compared to VDT_C or VDT_S (Table 3), indicating that LS and SCM are shortened in the forward-bent postures (VDT_C and VDT_S) and are released in the neutral posture (VDT_U). This may be because the forward-bent posture causes upper cervical spine extension and anterior elevation of the scapula, shortening the LS and SCM and increasing muscle thickness and activity [7,44,45].

Tone_R_SM, Tone_L_SM, Stiffness_R_SM, and Stiffness_L_SM significantly decreased in VDT_S (Table 3). The SM for the head erector shows hypertonicity in the slouching-head forward posture, causing chronic dysfunction of the neck and increasing tension-type stress and headaches [46,47]. In this study, supporting the head weight using a traction device effectively reduced the hypertonicity of the SM. This effect of VDT_S will provide even more benefits, especially for exaggerated poor posture due to tablet or pad work, which is known to be difficult to improve [30,48]. 

On the contrary, there were no significant differences in the UT mechanical properties. Similar to previous studies, the tone and stiffness of the UT did not change in forward head posture [8,49]. This suggests that the arm position of fixed subjects on the desk can result in decreased muscular activity and stiffness.

### 4.4. Blood Velocity

A significant increase in the right CCA PSV in VDT_U was observed (Table 3), suggesting that the blood velocity change was induced by the released SCM in VDT_U. The SCM is placed anterolateral to the CCA, so the volume of the SCM can affect CCA blood velocity. According to Gill (2019), as the volume of the SCM increases, the blood velocity of the internal carotid artery decreases [50]. In this study, Stiffness_R_SCM decreased in VDT_U, possibly affecting R_PSV. Furthermore, because CCA is highly associated with brain activity [51], the brain activity caused by VDT_U should be investigated in future studies. 

### 4.5. Preference and Discomfort

The greatest discomfort was observed in VDT_C (Table 4). Although VDT_C was more familiar to VDT users, VADS was significantly low owing to physical and physiological benefits in VDT_S and VDT_U. In particular, VDT_S showed the lowest VADS, indicating that the tension of the SM may be a major factor in increasing VDT user discomfort. According to the previous study that an upright posture was not the one optimal correct posture, comfortable and relaxed posture may be safer and helpful than an upright posture for symptom relief [52]. Therefore, muscle relaxation with VDT_S and VDT_U can help reduce the discomfort for VDT users so that it is thought to be helpful in improving comfort at rest. 

### 4.6. Limitations

This study investigated the effect of three different workstations, but it was still limited to explain the long-term effect because the intervention was a single trial and the application time of each workstation was slightly short. In addition, because we recruited small, painless samples, it was difficult to emphasize specifically whether the three workstations were effective for all VDT users or patients with musculoskeletal disorders. Additionally, since the study evaluated only the SM, UT, LS and SCM, the changes in other muscles were not clear. Furthermore, the effect of head support was difficult to apply every VDT user because it was applied only to the forward head posture. Therefore, we still need to find out optimal method of applying head support. In further studies, application of head supports to various VDT postures like backward lean or upright posture should be conducted and long-term effects should be analyzed in the prolonged workstation environment. In addition, it needs to be expanded to clinical applications, including the patients with musculoskeletal disorders or disc degeneration. Finally, the use of traction device workstation should be investigated to prevent or relieve pain. Therefore, the proposed method can help prevent cumulative trauma disorders such as disc degeneration and myalgia, which are commonly induced by a long-term bad posture in VDT users, thereby providing high-quality and healthy work life. 

## 5. Conclusions

This study demonstrated the cognitive and physical problems caused by the conventional VDT posture and examined the effects of the traction device workstation, proving its effectiveness.

Forward head posture in VDT_C increased musculoskeletal load, induced high stiffness or tone in the LS, SM, and SCM, and responded late during working memory tasks. In VDT_S, however, the benefits of head weight reduction provided better comfort and reduced SM tone and stiffness, but there was no difference in working memory performance compared with VDT_C. In VDT_U, the traction device stably maintained upright alignment during VDT work in the normal CVA range for a few minutes. The tone and stiffness of the LS and SCM significantly decreased, but the R_PSV increased. In particular, the working memory task performance is improved (faster RT).

In conclusion, VDT_S influenced head weight reduction, but maintaining the neutral alignment of VDT_U was more effective in improving working memory performance than simply reducing the musculoskeletal load. Therefore, being in an upright posture during work helps to improve the musculoskeletal safety and/or work productivity, and the traction device has positive effects on being upright, muscle tension relief, blood circulation and comfort. Therefore, we expect it to be a practical solution for large-scale VDT users.

## Figures and Tables

**Figure 1 ijerph-19-14254-f001:**
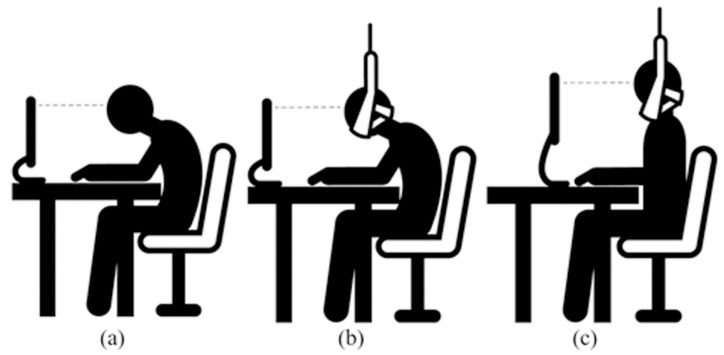
The three different kinds of visual display terminal (VDT) workstations. (**a**) Conventional VDT workstation (VDT_C); (**b**) Support VDT workstation (VDT_S); (**c**) Upright VDT workstation.

**Figure 2 ijerph-19-14254-f002:**
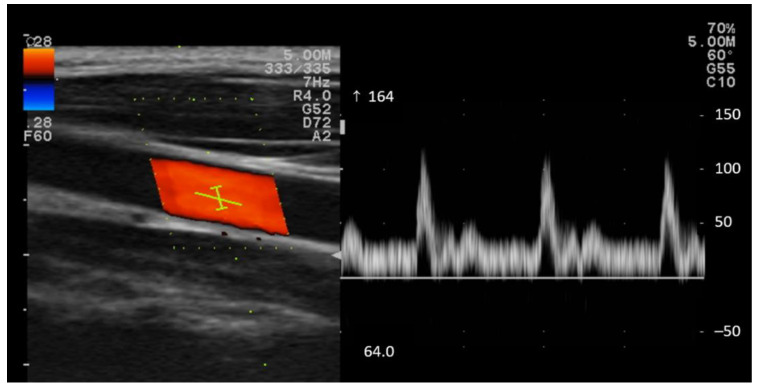
Common carotid artery (CCA) color Doppler waveform. The arrow (↑) represents a positive direction and the maximum displayable velocity value, such as 164 cm/s.

**Table 1 ijerph-19-14254-t001:** General characteristics of participants.

	Total (*n* = 34)	Men (*n* = 18)	Women (*n* = 16)
Age (years)	22.29 ± 2.02	22.72 ± 2.23	21.81 ± 1.63
Height (cm)	170.32 ± 7.72	176.19 ± 5.25	163.27 ± 2.65
Weight (kg)	67.59 ± 11.65	74.19 ± 10.07	60.17 ± 8.38
CVA (°)	39.13 ± 5.86	36.74 ± 5.66	41.82 ± 4.83
VDT usage time per day (hour)	8.24 ± 3.25	7.50 ± 3.15	9.06 ± 3.17

Abbreviation: CVA, cranio-vertebral angle; VDT, visual display terminal.

**Table 2 ijerph-19-14254-t002:** Repeated measure analysis results of 2-back task reaction time.

Repeated Measure Comparisons					Post Hoc Comparisons (Tukey)			
Outcome Variable	Variables	Mean ± SD	*F*	*p*	Variables		*t*	*p*
	VDT_C	673.93 ± 212.21			VDT_C	VDT_S	−0.65	0.796
RT (ms)	VDT_S	683.20 ± 233.68	5.32	0.007 *		VDT_U	2.64	0.033 *
	VDT_U	640.45 ± 179.69			VDT_S	VDT_U	2.52	0.044 *

Abbreviations: RT, reaction time; VDT_C, conventional visual display terminal workstation; VDT_S, support visual display terminal workstation; VDT_U, upright visual display terminal workstation; SD, standard deviation. * Statistically significant difference: *p* < 0.05.

**Table 3 ijerph-19-14254-t003:** Repeated measure analysis results of quantitative outcome variables.

Repeated Measure Comparisons					Post Hoc Comparisons (Tukey)			
Outcome Variable	Variables	Mean ± SD	*F*	*p*	Variables		*t*	*p*
	VDT_C	39.13 ± 5.86			VDT_C	VDT_S	−2.75	0.026 *
CVA (°)	VDT_S	41.63 ± 7.00	115.30	<0.001 *		VDT_U	−13.72	<0.001 *
	VDT_U	54.52 ± 4.32			VDT_S	VDT_U	−10.53	< 0.001 *
	VDT_C	17.87 ± 1.70			VDT_C	VDT_S	−2.07	0.112
Tone_R_LS (Hz)	VDT_S	18.39 ± 1.57	9.02	<0.001 *		VDT_U	2.36	0.062
	VDT_U	17.24 ± 1.50			VDT_S	VDT_U	3.91	0.001 *
	VDT_C	18.08 ± 1.70			VDT_C	VDT_S	−3.97	0.001 *
Tone_L_LS (Hz)	VDT_S	18.87 ± 1.52	20.41	<0.001 *		VDT_U	2.73	0.027 *
	VDT_U	17.48 ± 1.90			VDT_S	VDT_U	6.01	<0 .001 *
	VDT_C	17.20 ± 1.45			VDT_C	VDT_S	4.55	<0.001 *
Tone_R_SM (Hz)	VDT_S	16.14 ± 1.56	7.53	0.001 *		VDT_U	1.29	0.413
	VDT_U	16.79 ± 1.78			VDT_S	VDT_U	−2.41	0.055
	VDT_C	17.40 ± 1.64			VDT_C	VDT_S	2.91	0.017 *
Tone_L_SM (Hz)	VDT_S	16.59 ± 1.71	3.92	0.025 *		VDT_U	0.92	0.634
	VDT_U	17.07 ± 1.71			VDT_S	VDT_U	−2.16	0.094
	VDT_C	343.82 ± 60.17			VDT_C	VDT_S	−4.23	<0.001 *
Stiffness_L_LS (N/m)	VDT_S	368.97 ± 56.87	11.36	<0.001 *		VDT_U	1.23	0.445
	VDT_U	334.88 ± 66.83			VDT_S	VDT_U	4.11	<0.001 *
	VDT_C	324.79 ± 44.97			VDT_C	VDT_S	3.77	0.002 *
Stiffness_R_SM (N/m)	VDT_S	296.09 ± 46.84	6.08	0.004 *		VDT_U	0.24	0.968
	VDT_U	322.24 ± 54.33			VDT_S	VDT_U	−2.93	0.016 *
	VDT_C	328.32 ± 48.63			VDT_C	VDT_S	3.07	0.012 *
Stiffness_L_SM (N/m)	VDT_S	302.91 ± 56.25	5.28	0.007 *		VDT_U	0.40	0.916
	VDT_U	324.50 ± 51.73			VDT_S	VDT_U	−2.95	0.016 *
	VDT_C	211.68 ± 27.52			VDT_C	VDT_S	0.63	0.805
Stiffness_R_SCM (N/m)	VDT_S	208.53 ± 31.00	3.48	0.037 *		VDT_U	2.95	0.016 *
	VDT_U	200.29 ± 19.43			VDT_S	VDT_U	1.86	0.168
	VDT_C	113.57 ± 16.86			VDT_C	VDT_S	1.95	0.142
R_PSV (cm/s)	VDT_S	109.82 ± 16.37	4.80	0.012 *		VDT_U	−1.47	0.319
	VDT_U	116.40 ± 20.81			VDT_S	VDT_U	−2.71	0.029 *

Abbreviations: CVA, craniovertebral angle; R, right; L, left; LS, levator scapulae; SM, suboccipital muscles; SCM, sternocleidomastoid; RT, reaction time; PSV, peak systolic velocity; VDT_C, conventional visual display terminal workstation; VDT_S, support visual display terminal workstation; VDT_U, upright visual display terminal workstation; SD, standard deviation. * Statistically significant difference: *p* < 0.05.

**Table 4 ijerph-19-14254-t004:** Repeated measure analysis results of VADS.

Repeated Measure Comparisons					Post Hoc Comparisons (Tukey)			
Outcome Variable	Variables	Mean ± SD	*F*	*p*	Variables		*t*	*p*
	VDT_C	4.94 ± 2.00			VDT_C	VDT_S	3.29	0.007 *
VADS	VDT_S	3.82 ± 1.85	4.48	0.015 *		VDT_U	2.30	0.070
	VDT_U	4.00 ± 2.06			VDT_S	VDT_U	−0.39	0.918

Abbreviations: VADS, visual analogue discomfort scale; VDT_C, conventional visual display terminal workstation; VDT_S, support visual display terminal workstation; VDT_U, upright visual display terminal workstation; SD, standard deviation. * Statistically significant difference: *p* < 0.05.

## Data Availability

The data presented in this study are available upon request from the corresponding author.

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
