# Peer review of "Effects of a Traction Device for Head Weight Reduction and Neutral Alignment during Sedentary Visual Display Terminal (VDT) Work on Postural Alignment, Muscle Properties, Hemodynamics, Preference, and Working Memory Performance"

_ijerph, 2022, doi:10.3390/ijerph192114254_

Round 1

Reviewer 1 Report

Thanks for give me the possibility to review the current manuscript. In general, the topic of the article is very interesting, and the manuscript shows remarkable data and findings. Just minor things must be taken into consideration by the authors.

·      Line 21: info about de participants should be included

·      Line 67: reference about physical exercises required

·      Line 129: I consider it necessary to go into more detail on the concept of "ordinary posture".

·      Line 164: reference about CVA measurement required

·      Line 203: when the participants use the The Visual Analogue Discomfort Scale (VADS) exactly during the process?

·      Discussion: could be interesting discuss your results in comparison with similar studies (Zirek E, Mustafaoglu R, Yasaci Z, Griffiths MD. A systematic review of musculoskeletal complaints, symptoms, and pathologies related to mobile phone usage. Musculoskelet Sci Pract. 2020 Oct;49:102196. doi: 10.1016/j.msksp.2020.102196. Epub 2020 May 27. PMID: 32861360.)

·      I recommend including a section on limitations: muscle groups, reduced sample size and very specific characteristics, protocol application times, etc.

·      In general, posture concept is being redefines in the last years (Slater D, Korakakis V, O'Sullivan P, Nolan D, O'Sullivan K. "Sit Up Straight": Time to Re-evaluate. J Orthop Sports Phys Ther. 2019 Aug;49(8):562-564. doi: 10.2519/jospt.2019.0610. PMID: 31366294.). Should be interesting discuss about it.

Reviewer 2 Report

Dear Corresponding Author

thank you for submitting your paper. The issue is very interesting because it is very applicable for human daily life on a large scale of individuals. However, I want to stress some part of your paper, you can read in the following lines the details.

- From the paper I don't understand if the VDT-C position, as you draw in the figure 1, is maintained voluntary by the subject or it is the sum of the postural behavior during the time span. It is crucial to understand because in case it is voluntary the rest of the paper drops down. Please could ou provide me better information?

- the working memory paragraph is not clear at all. I mean, it is not clear which is the first and the second task. Please correct.

- lines 178-179: please can you add a reference or can you explain well why you give that kind of definition to muscle tone?

- in the conclusion I suggest to stress the practical applications of your results, suggesting to use the upright posture during working.

Regards

Round 2

Reviewer 2 Report

Thanks for the revised version of your paper.